# Discovering Correspondences between Multiple Languages by MDL

## Abstract

How can we automatically discover the most important correspondences between words from two or more languages? How can we do so allowing for correspondences between any subset of languages, without drowning in redundant results, and at the same time maintaining control over the level of detail? These are exactly the questions we answer in this paper.

We approach the problem with the Minimum Description Length principle, and give an efficient algorithm for discovering statistically important correspondences. We test the efficacy of our method against a set of Slavic languages. The experiments show our method automatically discovers non-trivial associations, allowing for both quantitative and qualitative analysis of multiple languages.

## 1  Introduction & Related Work

Systematic correspondences between languages form the basis for much linguistic work. Researchers employ them to e.g. improve teaching, analyze and quantify the similarity or relatedness of languages, or to formulate hypotheses about mutual intelligibility. Beyond that, they are of importance in multi-language natural language processing, notably in machine translation.

Correspondence rules can be established on the basis of various linguistic features, such as the language's alphabets, their orthographies, their phonologies, or their inflectional and derivational morphologies. As an example, the Czech, Polish, Russian, and Bulgarian forms of the pan-Slavic word for *happiness* could be analyzed to have the following ortho-phonetic correspondences:

| (PL) | szcz | ę | ści | e |
|------|------|---|-----|---|
| (CS) | št | ě | st | í |
| (RU) | сч | a | сть | e |
| (BG) | щ | a | ст | ие |

In order to find correspondences, a linguist typically collects cognates from two or more languages and compares them manually. If the linguist observes an often-occurring pattern, or one that fits well with other known changes that occurred between the languages, then she might conclude that this pattern is systematic and use it as basis for further investigations. This technique, called the *comparative method*, dates back to at least the 1800s (Szemerenyi, 1970).

Recently, researchers have devised various statistical approaches to identifying the regular correspondences between languages. Much of these focus on cognate identification or reconstruction (Schulz et al., 2004; Snyder et al., 2010), on discovering and quantifying etymological relationships between languages (Wettig et al., 2011), or on discovery of pseudo-morphological sub-word alignments (Snyder and Barzilay, 2008).

However, most existing statistical approaches are afflicted by a number of problems: they may impose arbitrary assumptions on the distribution or shape of correspondences, may not allow for integration of linguistic knowledge, or may be limited to pairs of languages. While imposing assumptions is sometimes necessary in order to obtain any results at all, it leads to finding not the "true" correspondences hidden in the data, but their closest similes from the assumed distribution.

How, then, can we discover correspondences between more than two languages without prior assumptions about their shape or distribution? This is the question we answer in this paper. For this, we employ the Minimum Description Length (MDL) principle (Grünwald, 2007). MDL provides a statistically well-founded approach to identifying the

*best* model for given data, and has amongst others been used to model changes in etymologically-related words (Wettig et al., 2011).

Using MDL, we deem the set of correspondences that describes the data most *succinctly* to be the best. We propose an efficient, deterministic algorithm to infer good sets of correspondence rules directly from data. In our experiments, we present a phylogenetic analysis of a number of Slavic languages and show how our approach can be used for efficient, highly detailed quantification of string-level similarities among more than two languages.

In our pairwise analysis, we confirm that string-level similarity between languages is a strong reflection of linguistic classification. In our four-language experiment, we find that our algorithm successfully identifies linguistic sub-groups while quantifying the similarity between all subsets of our four analyzed languages.

The paper is structured as follows: we give an overview of our approach and terminology in Section 2, then present our model in Section 4. After this, we report learned correspondences and provide information-theoretic analyses of language similarity in Section 5, and conclude in Section 6.

## 2 Approach & Terminology

We seek models which consist of sets of correspondence rules. We propose that correspondences should be treated simply as associated strings of characters with no assumed underlying distribution. Doing this results in objective, unbiased string-level measures of linguistic similarity and allows to observe the actual distributions of correspondence rules. Furthermore, we want to learn our rules deterministically, to exclude chance's influence on our assessments.

We build our approach on these key principles and use a number of observations to realize them.

1) In order to evaluate how good a given set of correspondence rules is, we should evaluate how well these rules describe the data. However, it is not immediately obvious how to do this. For example, if we are given the Polish-Czech correspondences (s,š), (sz,š), (c,t), (cz,t), and (szcz,št), then we can segment, or *align*, the initial sub-strings *szcz* and *št* from our *happiness* example in multiple ways. Three possible alignments are:

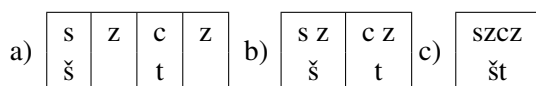

Lacking an evaluation function, we cannot tell which of the three example alignments above is the best. However, if we are given the best alignment of our data, then we can straightforwardly compute probabilities for each of the correspondence rules, from which we can then compute the optimal rule costs. Similarly, knowing the costs of the rules allows us to compute the optimal alignment.

Thus, our problem lends itself well to an Expectation-Maximization (EM) (Dempster et al., 1977) approach.

To use EM, we must formulate both the expectation and the maximization steps. The expectation step is straightforward; we simply align the data with the current model. The maximization step can be made intuitive with another observation.

2) If the optimal alignment is c), using rule (szcz,št), then *any* of the rules making up alignments a) and b) occur at least as often in the data as (szcz,št) does. Therefore, improving the model can be done by checking if an alignment is improved by *merging any two compatible rules*.

3) With this in mind, initialization of this model is straightforward: if we start by assuming no structure at all, then we will find the dominant structures in the data. Therefore, we start training from what we call a *null alignment*: one which uses only rules which contain exactly one symbol from exactly one language. An example for such a rule is (z,) in alignment a). We call such correspondence rules *singleton* rules.

We next explain how we formalize our model in terms of MDL.

## 3 MDL for Correspondence Discovery

The Minimum Description Length principle proposes that the optimal model is the one resulting in the most *concise* description of the modeled data. Importantly, in MDL-based modeling, we use only the data as evidence for our models and forego making assumptions about the nature of models in the form of prior probabilities. This makes it ideally suited for our purposes.

For our model, we employ two-part MDL (Grünwald, 2007), which consists of a stack of two-part codes. The main formula for this is

$$\mathbf{M} = \arg\min_{M \in \mathcal{M}} L(M) + L(D|M),$$

with $D$ being the data at hand, $M$ the explaining model, and $\mathcal{M}$ the *model class* we draw our models

from. $L(M)$ is the length, in bits[1], of the description of model $M$. Similarly, $L(D|M)$ is the length, in bits, of the data given model $M$. Description lengths are simply code lengths: Shannon's source coding theorem (Shannon, 2001) tells us that the best prefix-free code for some data is derived from the (negative logarithm of the) probabilities of the data, i.e.

$$L(M) + L(D|M) = -\log p(M) - \log p(D|M).$$

From this, we see that two-part MDL can be considered to be a regularized maximum likelihood approach very similar to Bayesian inference. A crucial difference is that in MDL, $L(M)$ serves as statistical formalization of the desired model *class*, whereas in Bayesian inference the analogous term expresses prior beliefs about the distribution of model *parameters*.

Due to MDL's roots in coding theory, it is common to call an MDL objective function a *code*, and to speak of *encoding*, *transmitting*, or *sending* the individual elements that make up a code.

We next detail how we design our code.

## 4 The PS-2 MDL Code

Given a list $D$ of cognate tuples from $N$ languages, we use the Minimum Description Length principle to infer the statistically significant correspondence rules. Our encoding builds upon and extends the encoding introduced by (Tatti and Vreeken, 2012) for discovering small sets of serial episodes in event sequences.

We proceed with a two-part code, requiring us to solve the optimization problem

$$\mathbf{M} = \arg\min_{M \in \mathcal{M}} L(M) + L(D|M).$$

First, we must determine the model class, $\mathcal{M}$. As explained in Section 2, we here seek to find associated character strings between languages. Thus, mathematically our model class is the set of sets of tuples associating strings from the individual languages' alphabets. We make no assumptions about either their shape or their distribution.

We begin by discussing our model code $L(M)$.

### 4.1 Model Code L(M)

Our models consist of $N$ alphabets $\Sigma_i$ and a correspondence rule table which we call $\Pi$. Our total

---
[1]We use $\log(.) = \log_2(.)$ throughout the paper.

model description length is given by

$$L(M) = \sum_{i=1}^{N} (L(\Sigma_i)) + L(\Pi).$$

In order to describe the rules from $\Pi$, we require the code lengths for all letters $\sigma$ from all alphabets $\Sigma_i$. Since we are interested only in complexities, we disregard the actual code words and focus only on their lengths. For ease of exposition, we first discuss $L(\Pi)$.

In essence, our $\Pi$ is a list of independent correspondence rules. We next describe how we encode an individual rule.

**Encoding a Correspondence Rule** Rules $\pi \in \Pi$ are of the form $\pi = (\pi_1, ..., \pi_N)$ with $\pi_i \in \Sigma_i^*$. To encode one such rule, we must specify a) how long the string from each of the languages is and b) which letters it contains.

However, if we specify lengths even when they are zero, we pay with higher description lengths than are necessary. This imposes a bias particularly against rules which are sparsely populated. Thus, we encode each rule $\pi \in \Pi$, $\pi = (\pi_1, ..., \pi_N)$ as follows: we transmit all $N$ entries independently of each other, sending their lengths and character sequences only where defined:

$$L(\pi) = \sum_{\substack{n=1 \\ \pi_n \neq \epsilon}}^{N} \left( L_{\mathbb{N}}(|\pi_n|) + \sum_{\sigma \in \pi_n} L(\text{code}(\sigma)) \right).$$

We encode each string's length with $L_{\mathbb{N}}$, the *universal code for the integers* (Rissanen, 1983), which is the MDL-optimal code for natural numbers of unknown, arbitrary size. For transmitting the strings itself, we use $\text{code}(\sigma)$, i.e. the optimal unigram for symbol usages in all rules.

**Encoding the Rule Table** If we want to use the code for individual rules given above, we must specify for every rule which subset of languages it is defined on. We can straightforwardly classify each rule according to which subset it is defined on. Then, we must specify how many rules defined on each of the different subsets there are.

There are $2^N - 1$ different language subsets on which a rule may be defined. We encode the number of rules of each kind via $L_{\mathbb{N}}$. These numbers must be offset since $L_{\mathbb{N}}(n)$ is defined for $n \geq 1$, and there may be zero rules of a certain kind.

Thus:

$$L(\Pi) = \sum_{i=1}^{2^N-1} \left( L_\mathbb{N}(|\Pi_{C_i}| + 1) + \sum_{\pi \in \Pi_{C_i}} L(\pi) \right)$$
$$+ L_\mathbb{N}(T_\Pi) + \log\left(\binom{T_\Pi - 1}{|\Pi| - 1}\right).$$

where $\text{count}(\pi)$ is the number of occurrences of $\pi$, $T_\Pi = \sum_{\pi \in \Pi} \text{count}(\pi)$ and where $\Pi_{C_i}$ is the set of rules defined on the $i$-th subset of languages enumerated in some canonical way.

Additionally, to describe our data items using rules, we must specify the optimal code lengths for using each of the rules. In $L(\Pi)$, this corresponds to the last two summands. We do this by a *data-to-model code* (Grünwald, 2007). Data-to-model codes are used to code uniformly from an enumeration of models, i.e. without preference towards any particular model. Since we know that none of the symbols of any alphabets will have a non-zero count, the data-to-model code is given by the weak number composition of the alphabets symbols' total counts over the number of symbols.

**Encoding the Alphabets** For describing the strings of each rule, we use the Shannon-optimal code for the individual alphabets' symbols. Thus, we must first transmit the unigrams, i.e. $\text{code}(\sigma)$ $\forall \sigma \in \Sigma_i$ for every alphabet $\Sigma_i$. We again do this by a data-to-model code, i.e. by coding uniformly from all possible distributions.

Setting $T_{\Sigma_i} = \sum_{\sigma \in \Sigma_i} \text{count}(\sigma)$, the total transmission cost relating to some $\Sigma_i$ becomes

$$L(\Sigma_i) = L_\mathbb{N}(T_{\Sigma_i}) + \log\left(\binom{T_{\Sigma_i} - 1}{|\Sigma_i| - 1}\right).$$

The alphabet sizes are constant for any given data set and therefore it is not necessary to include them in the code.

Having described our model, we next turn our attention to encoding the data with a given model.

### 4.2 Data Code L(D|M)

To encode data with our model, we simply transmit the correspondences best used to describe each data entry. Thus, we get

$$L(D|M) = \sum_{d \in D} L(d|M)$$

where $L(d|M) = L_\mathbb{N}(|d|) + \sum_{\pi \in d} L(\text{code}(\pi))$.

Again, it is not necessary to specify the number of data entries as they do not change for the same data set. For the individual data entries, we transmit their lengths via $L_\mathbb{N}$ and specify which correspondence rules they are best aligned with via the best usage code for the rules, $\text{code}(\pi)$.

This leaves us to discuss finding the data's description given some rules, and how to infer rules.

### 4.3 Alignment Procedure

Computationally, finding the best description for a data item boils down to finding the best alignment for it. We formulate this as a shortest-path problem in a weighted, directed graph and use Dijkstra's algorithm (Dijkstra, 1959) to find optimal alignments. Nodes in the graph represent index tuples, while edges describe the applicable rules.

By partial order reduction, we make our graphs as small as possible. Nonetheless, due to the combinatorial nature of the problem, there are bottlenecks in memory consumption as well as in runtime. With our current implementation, we can process up to five languages from our most complicated data set within a few hours on a 4GB RAM, 2.5GHz single core desktop machine. We plan to extend this to higher $N$ in future work.

### 4.4 Training Procedure

Inferring correspondences of arbitrary length is a combinatorial, non-convex optimization problem defined over a large, unstructured search space. However, as we argued in Section 2, if we are given a rule table with costs, we can compute the optimal alignment of all data with these rules. Likewise, if we are given an alignment of all data, we can improve our model from it. Therefore, we can find good solutions by Expectation-Maximization (Dempster et al., 1977).

**Initialization** At the beginning of training, we either initialize our model with a *null alignment* or with a greedy alignment from a given set of rules. A null alignment is one in which only *singleton rules* are used, i.e. only rules which consist of exactly one character from exactly one language. Starting from a given rule set allows to input linguistic knowledge in an intuitive way.

**Expectation Step** In the Expectation step, we align all data items with the rules from the current rule table $\Pi$ and the current usage costs for the rules. This results in new counts for all rules from which we compute costs in the next step. We

employ Laplace correction in order to ensure that the algorithm is always able to explain all data and may choose to not use locally suboptimal patterns.

The time complexity of our E step is in $\mathcal{O}(|D| \cdot |R|^2)$, where $R$ is the maximum number of possible rule applications in a single data entry.

**Maximization Step**  In the Maximization step, we optimize our code table. We do this by merging together the two patterns which lead to the highest decrease in overall description length. The intuition behind this is the observation that if a longer pattern is useful, then any sub-pattern of it will be at least as or more useful. It is important to note that in this way, the learned correspondences *grow according to their statistical significance.*

Each M step has time complexity $\mathcal{O}(|D| \cdot A^2/2)$, where $A$ is the maximum number of rules used to align a single data entry.

It is possible that a rule $R$ is deemed good on the basis of the entire data set when in fact it is suboptimal for some subset $S$ of the data. Let us assume that the entries in $S$ have become aligned with rule $R$ even though an overlapping, but different rule $Q$ would have been a better choice. Then, $S$ will be "lost" in regards to discovering $Q$, as $S$ will not be counted as evidence for $Q$. However, if the remaining data contain enough evidence to learn the rule $Q$ independently of $S$, $Q$ will be in fact learned and may then be used for aligning $S$.

In this fashion, we deterministically learn the important structures in the data, although occasionally we may miss some of the more subtle correspondence rules.

### 4.5 Data Over-Weighting

MDL guards against overfitting by balancing the complexity of the model with that of the data – only those correspondences with sufficient evidence in the data are included. By confining ourselves fully to the data and not relying on any further assumptions, we obtain objective results. However, it may be that our focus in on the obtained rules rather than on the objective statistical analysis of the data.

For example, in cognate reconstruction, we require rules which relate strings across languages, but not those which exclusively describe substrings from the separate languages – the latter contribute nothing to successfully adapting a word from its known form(s) to an unknown one.

We can also encounter problems working on under-resourced or very rich languages, or even

when simply exploring correspondences between higher numbers of languages, where the need for data may exceed our ability to provide it. Sometimes, we simply may not have sufficient statistical evidence to discover all desired rules.

In such cases we can turn to correspondences which objectively are not statistically significant for the given data, but are almost so. We can straightforwardly include those near-significant correspondences by over-representing the data. At the corpus level, this can be done by simply assigning the data complexity term a higher weight than the model term, i.e. via optimizing

$$L(M) + \alpha L(D|M)$$

with $\alpha \neq 1$ instead of the original formula. In this formula, a data weight of e.g. 2 corresponds to using twice the amount of identical data. More subtly, we can also assign individual cognate tuples a higher weight than others. Again, assigning some data entry a weight of 2 intuitively duplicates that entry in the data set. With the data over-represented in such a fashion, the algorithm will have enough evidence to include larger correspondence rules than before, as they effectively become more useful for describing the (over-represented) data. Nonetheless, new rules will be discovered in the "correct" order; that is, ordered by statistical significance.

Clearly, this moves results away from the objective territory of MDL and introduces a user's subjective judgment. However, it allows a linguistic expert to choose a desired level of detail for inferred correspondences.

We illustrate the effects and helpfulness of this idea experimentally, in Section 5.

## 5 Experiments & Results

We show our approach's efficacy in several ways.

First, we present a standard pairwise analysis for a group of languages for which we have collected data we deem representative. We compute pairwise distances, construct a phylogenetic tree, and compare this tree to linguistic classifications.

Second, we present a detailed analysis of four languages simultaneously. Because our approach is not limited to pairs of languages, we can give an information-theoretic quantification of linguistic similarity in a much more detailed fashion than was previously possible.

In both cases, we report some of the learned correspondences. We also show the effect of data over-weighting, as introduced in Section 4.5.

We first discuss our data, then present results.

### 5.1 Data Sets

We compiled two data sets for our experiments. Firstly, we use Swadesh lists for 13 modern Slavic languages taken from the wiktionary.[2] The languages are Czech, Polish, Slovak, Lower Sorbian, Upper Sorbian (west Slavic), Russian, Belarussian, Ukrainian, Rusyn (east Slavic), Bulgarian, Macedonian, Slovenian, and Serbo-Croatian (south Slavic). For Serbo-Croatian, we have both a version in Latin script and one in Cyrillic script.

Secondly, we add a set of Slavic cognates containing internationalisms and pan-Slavic words for Czech, Polish, Russian, and Bulgarian.[3]

All our data is in raw orthographic form, without transcriptions of any kind. It consists mostly of verbs, adjectives, and nouns.

| data | all lang. | RU-BG | CS-PL | CS-PL-RU-BG |
|---|---|---|---|---|
| size | 207 | 778 | 778 | 778 |

TABLE 1: Data set sizes for experiments.

For all of our experiments, we use only those entries that contain words for all languages in question. While our algorithm is agnostic to gaps in data, this makes for easier comparison.

### 5.2 Pairwise Analyses

For pairwise analysis, we require some measure of distance between pairs of languages. In MDL-based modeling, it is common to use *Normalized Compression Distance* (NCD) (Cilibrasi and Vitanyi, 2005) for this. Intuitively, NCD measures how hard it is to describe $X$ and $Y$ together compared to how hard it is to describe them separately. It is defined as

$$NCD(X,Y) = \frac{L(X,Y) - \min(L(X,X), L(Y,Y))}{\max(L(X,X), L(Y,Y))}$$

where $L(X,Y)$ is the description length when encoding languages $X$ and $Y$ jointly. NCD is a mathematical distance; lower values mean that two data sets are more similar.

We train models for all pairs from our languages and obtain correspondence rules and NCDs.

---

[2]Taken from https://en.wiktionary.org/wiki/Appendix:Slavic_Swadesh_lists.

[3]Compiled from (Likomanova, 2004) and (Angelov, 2004).

**NCDs** In Table 2 we show the NCD values for all pairwise comparisons. We use ISO 639-1 and ISO 639-3 codes to identify the languages, except for Serbo-Croatian, which we denote by $SC_l$ in its Latin version and and $SC_c$ in its Cyrillic version. We indicate **lowest** and *highest* NCDs per row in bold and italic text, respectively.

Our table reveals that languages from the same linguistic group tend to have lower NCD than languages from differing groups. The south Slavic group is linguistically further divided into a southwestern group (Slovene and Serbo-Croatian) and a southeastern sub-group (Macedonian and Bulgarian). Indeed we identify Slovenian and Serbo-Croatian as more similar to languages from the west Slavic group than to the east Slavic group. We can see that the Serbo-Croatian data in Latin script was assessed to be slightly closer to the other languages that use Latin script, while the Cyrillic version was deemed more similar to other languages using Cyrillic.

| | usb | lsb | CS | SK | PL | SL | $SC_l$ | $SC_c$ | MK | BG | RU | UK | rue | BE |
|---|---|---|---|---|---|---|---|---|---|---|---|---|---|---|
| usb | .00 | **.52** | .53 | .52 | .60 | .57 | .61 | .62 | *.76* | .75 | .68 | .70 | .67 | .64 |
| lsb | **.52** | .00 | .65 | .66 | .72 | .67 | .68 | .71 | *.87* | .85 | .80 | .82 | .78 | .74 |
| CS | .53 | .65 | .00 | **.41** | .56 | .50 | .53 | .55 | *.71* | .69 | .61 | .64 | .58 | .59 |
| SK | .52 | .66 | **.41** | .00 | .58 | .48 | .51 | .56 | *.68* | .66 | .60 | .65 | .59 | .60 |
| PL | .60 | .72 | **.56** | .58 | .00 | .64 | .64 | .67 | *.82* | .79 | .71 | .74 | .69 | .63 |
| SL | .57 | *.67* | .50 | .48 | .64 | .00 | **.36** | .39 | .59 | .58 | .61 | .65 | .60 | .61 |
| $SC_l$ | .61 | *.68* | .53 | .51 | .64 | .36 | .00 | **.04** | .54 | .57 | .63 | .66 | .62 | .63 |
| $SC_c$ | .62 | *.71* | .55 | .56 | .67 | .39 | **.04** | .00 | .51 | .53 | .60 | .63 | .59 | .59 |
| MK | .76 | *.87* | .71 | .68 | .82 | .59 | .54 | **.51** | .00 | .54 | .74 | .78 | .75 | .75 |
| BG | .75 | *.85* | .69 | .66 | .79 | .58 | .57 | **.53** | .54 | .00 | .70 | .77 | .70 | .71 |
| RU | .68 | *.80* | .61 | .60 | .71 | .61 | .63 | .60 | .74 | .70 | .00 | .52 | .53 | **.51** |
| UK | .70 | *.82* | .64 | .65 | .74 | .65 | .66 | .63 | .78 | .77 | .52 | .00 | .45 | **.45** |
| rue | .67 | *.78* | .58 | .59 | .69 | .60 | .62 | .59 | .75 | .70 | .53 | **.45** | .00 | .54 |
| BE | .64 | *.74* | .59 | .60 | .63 | .61 | .63 | .59 | .75 | .71 | .51 | **.45** | .54 | .00 |

TABLE 2: NCDs for 13 Slavic languages.

**Inferred Phylogenetic Tree** For easier viewing, we construct a phylogenetic tree from the NCD values, which we show in Figure 1. For this we use the neighbor joining method (Saitou and Nei, 1987) and place the root manually.[4] The greater the horizontal distance between two languages, the less similar they are.

As we see, the algorithm groups the languages according to their linguistic classification. It identifies Bulgarian and Macedonian as slight outliers

---

[4]Picture generated with http://etetoolkit.org/treeview/, tree generated with scikit-bio: http://scikit-bio.org/.

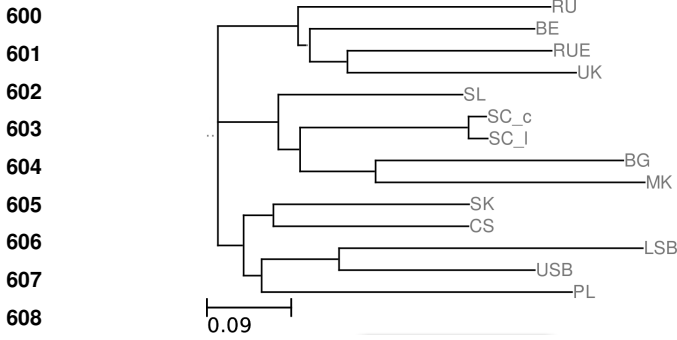

FIGURE 1: NCD-based Slavic phylogenetic tree.

in the south Slavic group, and Polish, Upper and Lower Sorbian as such in the west Slavic group.

This is an expected result. Bulgarian and Macedonian are outliers in that they have largely lost case declension. Adjectives and verbs from these languages oftentimes employ zero endings or comparatively shorter endings than the other languages. This leads to overall lower BG-BG and MK-MK description lengths, but still high NCDs to the other languages. In the Swadesh list, Upper and Lower Sorbian words are often of a different etymological heritage than words from the other Slavic languages. This explains their outlier status. Polish is a slight outlier due to its frequent use of digraphs and prolific palatalization, which increases complexity of Polish patterns.

**Inferred Correspondences** In Table 3, we present some example alignments from the models for the CS-PL and RU-BG language pairs.

To give insight into what kinds of rules emerge naturally, and what kind emerge from overrepresenting data as proposed in Section 4.5, we present two different alignments per selected example: firstly one obtained from an objective model, and secondly one obtained from overrepresented data. For the latter, we over-weight our data until there are no more singleton rules in any of the alignments. We mark over-weighted versions with *.

| (PL) | z | e | m | ě | | *ziem | ia | | (PL) | m | i | l | c | z | | e | ć | | *mi | l | cz | eć |
|------|---|---|---|---|---|-------|----|---|------|---|---|---|---|---|---|---|---|---|-----|---|----|----|
| (CS) | z | ie | m | i | a | *zem | ě | | (CS) | m | | l | | | | č | et | | *m | l | č | et |
| (PL) | ró | | g | | *róg | (PL) | r | o | z | d | z | i | e | | l | i | ć | | *roz | dzie | li | ć |
| (CS) | r | | o | h | | *ro | h | | (CS) | r | o | z | d | | | | ě | l | i | t | | *roz | dě | li | t |

| (RU) | м | о | л | о | | д | о | с | т | ь | | *м | оло | до | сть |
|------|---|---|---|---|---|---|---|---|---|---|---|-----|-----|-----|-----|
| (BG) | м | | л | | а | д | о | с | т | | | *м | ла | до | ст |

| (RU) | п | ъ | л | ен | | | *пъл | ен |
|------|---|---|---|----|---|---|------|-----|
| (BG) | п | о | л | | н | ый | *п | ол | ный |

TABLE 3: Example CS-PL, RU-BG correspondences.

As can be seen, the discovered correspondences are of different granularities and linguistic char-

acter. We find purely phonological rules such as (g,h) along with purely orthographic ones such as (cz,č), verb endings such as (ć,t), aberrations thereof such as (eć,et), liquid metatheses such as (ла,оло), palatalizations such as (dzie,dě), and even stem correspondences such as (ziem,zem).

Over-representing the data allows to select from a desired level of detail. We plan to discuss potential applications of the different resulting rules in future work.

## 5.3 Four-Way Analysis

Next, we turn our attention to more fine-grained analysis of linguistic similarity.

Restricting ourselves to pairwise analyses and grouping the *most* similar languages together in a phylogeny causes us to miss many subtle similarities. Existing approaches which are limited to pairwise correspondences incur infeasible amounts of computation when trying to use them for the simultaneous analysis of multiple languages.

Our algorithm is agnostic to the number of input languages and can be used to efficiently analyze more than two languages at a time. This allows for highly detailed information-theoretic quantification of the similarities among groups of languages. To show how this can be done, we first present some four-way CS-PL-RU-BG example alignments in Figure 4. They were computed **without** over-weighting.

| (PL) | p | i | | ć | | (PL) | s | p | e | | c | | j | a | l | | n | y | |
|------|---|---|---|---|---|------|---|---|---|---|---|---|---|---|---|---|---|---|---|
| (CS) | p | í | | t | | (CS) | s | p | e | c | | i | | á | l | | n | | í |
| (RU) | п | и | ть | | (RU) | с | п | е | ц | и | | и | аль | | н | ы | й |
| (BG) | п | и | я | | (BG) | с | п | е | ц | и | | а | л | | ен | |

| (PL) | m | i | ł | y | | (PL) | ś | m | i | | a | ł | y | |
|------|---|---|---|---|---|------|---|---|---|---|---|---|---|---|
| (CS) | m | i | l | ý | | (CS) | s | m | ě | | | l | ý | |
| (RU) | м | и | л | ый | | (RU) | с | м | | e | | л | ый | |
| (BG) | м | и | л | | (BG) | с | м | | e | | л | |

TABLE 4: Example CS-PL-RU-BG correspondences.

Observe that some of the discovered rules link only two or three languages, with the other language(s) described by separate patterns. We have selected some examples to highlight the differences in *i* vowels. In the data, there is enough evidence to discover various rules, such as (,,и,и), (,i,и,и), but not enough evidence to include a four-way rule (j,i,и,и). In consequence, the internationalism *specjalny* is analyzed with the three-way *i* correspondence plus a Polish singleton rule (j,,,). There are also cases where using two rules, such as (i,í,,) plus (,,и,и) in *pić* (*to drink*), is a good choice.

Because we designed our algorithm to discover the correspondence rules according to their statistical significance, this can be exploited for fine-grained analyses of similarity. Statistically, the observed correspondences between two or three languages are significant for the data, while potential larger ones are not. In other words: there is more regularity in some of the languages than others.

### 5.3.1 Shared Description Lengths

To quantify the amount of structure that individual languages share, we can compare the description lengths of their rules. For this, we define the **shared Description Length of languages** $L_{i_1}, ..., L_{i_k}$ as

$$sDL(L_{i_1}, ..., L_{i_k}) := \sum_{\pi \in Q(L_{i_1}, ..., L_{i_k})} L(\pi).$$

where $Q(L_{i_1}, ..., L_{i_k})$ contains all rules which are non-empty exactly for languages $L_{i_1}, ..., L_{i_k}$.

Figures 2 and 3 show sDLs for the CS-PL, RU-BG, and CS-PL-RU-BG models.

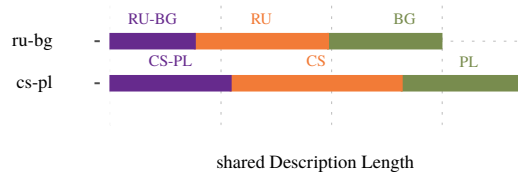

FIGURE 2: sDLs for the CS-PL and RU-BG models. Total rule desc. lengths: 1852.43 bits (CS-PL), 1496.62 bits (RU-BG)

Figure 2 reveals that RU diverges more than BG does from the RU-BG joint description, and that CS does so for CS-PL. Because we chose only cognate tuples defined for all four languages for this experiment, we can also compare the two language pairs. There, we see that CS-PL requires a larger description, and that Czech alone takes a somewhat larger fraction of total description length.

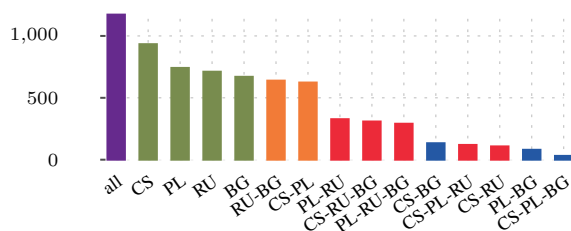

FIGURE 3: sDLs for the CS-PL-RU-BG model.

Figure 3 finally gives us a quantification of the similarities between all language sub-sets from our Czech-Polish-Russian-Bulgarian set. We see that the four-way sDL is the biggest contributor overall. It quantifies the complexity of the structure shared by all four languages.

We also see that each of the individual languages have significant overheads to the four-way shared description length. The language with the highest individual description length by far is Czech. This is not surprising, as Czech has the largest number of diacritically-modified symbols. For example, every Czech vowel can be marked as long with the *čárka*, giving us e.g. *é* as long version of *e*.

The algorithm furthermore identifies the linguistic grouping between Czech and Polish. Both CS-PL and RU-BG share significant portions of description length. In the NCD table, Table 2, we can observe that the south Slavic languages were somewhat in between the west and the east Slavic languages. In this analysis, we can see that Bulgarian is in fact very similar to Russian, so much so that in the four-way analysis, grouping RU-BG seems as good a choice as grouping CS-PL.[5]

Beyond this, we identify further, more subtle similarities. Taking a closer look we see they are between Russian and other, not-yet-covered language subsets. To highlight this, we have plotted the similarities to Russian in red. This is a highly satisfying result, as in fact we expect Russian, the Slavic language with the largest amount of native speakers, to heavily influence the other languages.

## 6 Conclusion

We studied the problem of automatically inferring objective string-level correspondences from data. We introduced an MDL-based approach and gave an efficient algorithm for finding correspondences.

Our experiments show that the approach works well in practice. We constructed a sensible phylogeny for our languages, demonstrated the discovered correspondences, and showed that our algorithm quantifies similarity not only between pairs, but between all subsets of analyzed languages.

While our algorithm is deterministic in its pure form, it is easy to integrate non-determinism e.g. by simulated annealing, since the alignments can be randomized in any step. However, the possibility to input linguistic knowledge and obtain deterministic results makes our approach particularly promising for linguists. We plan to discuss its linguistic potential in future work.

---

[5] Please note that this is purely on the basis of *superficial* word form similarity.

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
