# Peer review of "Discovering Correspondences between Multiple Languages by MDL"

_CoNLL 2016 — decision unknown_

[Official Review · Reviewer 1 · rating 3 · confidence 3]
soundness 4 · originality 4 · clarity 3 · impact 4 · substance 4 · appropriateness 5 · meaningful comparison 2 · replicability 3 · presentation format Oral Presentation

This paper proposes a method for discovering correspondences between languages
based on MDL. The author model correspondences between words sharing the same
meaning in a number of Slavic languages. They develop codes for rules that
match substrings in two or more languages and formulate an MDL objective that
balances the description of the model and the data given the model. 
The model is trained with EM and tested on a set of 13 Slavic languages. The
results are shown by several distance measures, a phylogenetic tree, and
example of found correspondences. 

The motivation and formulation of the approach makes sense. MDL seems like a
reasonable tool to attack the problem and the motivation for employing EM is
presented nicely. I must admit, though, that some of the derivations were not
entirely clear to me.
The authors point out the resemblance of the MDL objective to Bayesian
inference, and one thinks of the application of Bayesian inference in
(biological) phylogenetic inference, e.g. using the MrBayes tool. An empirical
comparison here could be insightful.  

Related work: 
- Lacking comparison to methods for borrowing and cognate detection or other
computational methods for historical linguistics. For example, the studies by
Alexandre Bouchard-Cote, Tandy Warnow, Luay Nakhleh and Andrew Kitchen. Some
may not have available tools to apply in the given dataset, but one can mention
List and Moran (2013). There are also relevant tools for biological phylogeny
inference that can be applied (paup, MrBayes, etc.). 

Approach and methodology
- Alignment procedure: the memory/runtime bottleneck appears to be a major
drawback, allowing the comparison of only 5 languages at most. As long as
multiple languages are involved, and phylogenetic trees, it would be
interesting to see more languages. I'm curious what ideas the authors have for
dealing with this issue. 
- Phylogenetic tree: using neighbor joining for creating phylogenetic trees is
known to have disadvantages (like having to specify the root manually). How
about more sophisticated methods?  
- Do you run EM until convergence or have some other stopping criterion? 

Data
- Two datasets are mixed, one of cognates and one not necessarily (the Swadesh
lists). Have you considered how this might impact the results? 
- The data is in orthographic form, which might hide many correspondences. This
is especially apparent in languages with different scripts. Therefore the
learned rules might indicate change of script more than real linguistic
correspondences. This seems like a shortcoming that could be avoided by working
on the level of phonetic transcriptions.

Unclear points
- What is the "optimal unigram for symbol usages in all rules"? (line 286)
- The merging done in the maximization step was not entirely clear to me. 

Minor issue
- "focus in on" -> "focus on" (line 440)

Refs
Johann-Mattis List, Steven Moran. 2013. An Open Source Toolkit for Quantitative
Historical Linguistics. Proceedings of the 51st Annual Meeting of the
Association for Computational Linguistics: System Demonstrations, pages
13â18, Sofia, Bulgaria. Association for Computational Linguistics.
http://www.aclweb.org/anthology/P13-4003.  
Andrew Kitchen, Christopher Ehret, Shiferaw Assefa and Connie J. Mulligan.
2009. Bayesian phylogenetic analysis of Semitic languages identifies an Early
Bronze Age origin of Semitic in the Near East